# What Drives the Choice of Local Seasonal Food? Analysis of the Importance of Different Key Motives

**DOI:** 10.3390/foods10112715

**Published:** 2021-11-06

**Authors:** Laura Maria Wallnoefer, Petra Riefler, Oliver Meixner

**Affiliations:** Department of Economics and Social Sciences, Institute of Marketing and Innovation, University of Natural Resources and Life Sciences, Vienna, 1180 Wien, Austria; laura.wallnoefer@boku.ac.at (L.M.W.); petra.riefler@boku.ac.at (P.R.)

**Keywords:** seasonal food, local food, choice-based conjoint analysis, discrete choice experiment, consumer preference, sustainable consumption, structural equation modeling

## Abstract

Local seasonal food choices are environmentally relevant behaviors and a promising opportunity for enhancing sustainable food consumption. Therefore, we need a more integrated understanding of motives driving consumers to opt for food that is produced locally and also in its natural growing season. The aim of this study is to (i) identify which motives for local food choices are also relevant for local seasonal food choices and (ii) investigate whether environmental motives become (more) relevant for these environmentally friendly choices. To assess consumer perceptions of socioeconomic, health, and environmental aspects, a survey in combination with a choice-based conjoint experiment to measure consumer preferences for seasonal (apples) and non-seasonal choices (tomatoes) was conducted. The data were collected by means of an online-panel survey (*n* = 499) and analyzed using two structural equation models. Results revealed that while the support of the local economy presents the most relevant driver, consumers’ price sensibility is even more relevant as a barrier. What differs is the relevance of authenticity and local identity. While local seasonal food provides environmental benefits to consumers, these benefits have no implications for the relevance of environmental motives. Based on these findings, we derive evidence-based recommendations for policymakers and marketers and propositions for future research regarding additional drivers and barriers for local seasonal food consumption.

## 1. Introduction

Food choices are environmentally significant behaviors linked to the exploitation of resources such as land, water, raw materials, and the emission of greenhouse gases (GHG) [1,2,3]. Globally, food consumption accounts for 48% of household impacts on land resources and 70% of impacts on water resources [2]. GHG emissions of non-vegetal foodstuffs mostly result from non-fossil emissions, whereas the emissions of vegetable foodstuffs mainly stem from energy use in farming, transportation, and preparation of food [4]. The almost constant availability of different food products, regardless of seasonal conditions, resulting from the globalization in the food trade, for example, has led to a remarkable increase in the travel distance of food [5]. Consequentially, consumers demanding food according to its place of origin, production process, or producers plays an important role in the sustainability discourse [6,7]. As such, individuals choosing to eat locally harvested, seasonal, and/or organic food and follow a vegetarian diet have a lower per capita environmental impact than those relying on more customary diets [8]. The transition towards sustainable diets basing on organic, local, and seasonal foods, thus, presents an opportunity to advance commitments to sustainable development [9,10]. Vita et al. [11], for example, recommend policies to favor the synergies between local, seasonal, and organic agriculture, as these might lead to dynamic effects that can further improve sustainable food consumption. To promote and implement relevant policies, knowledge about the individual’s motives driving the consumption of local, seasonal, and organic food products can be valuable. Such insights might assist policymakers and marketers in designing appropriate, target-group oriented communication strategies aiming at fostering sustainable food consumption [12,13]. Examples include public information campaigns and marketing cues aligned to underlying motives.

To provide relevant insights, researchers from different backgrounds investigated the role of different values, beliefs, and attitudes as drivers for sustainable food choices. Currently, the majority of studies focuses on either organic food [14,15,16], local food [5,17,18], or a combination of these two attributes [7,19,20,21]. In the case of local food, the review of Feldmann and Hamm [12] reveals that the consumer perceptions and preferences are manifold and relate to product quality (i.e., freshness, healthiness, and taste), the support of the local economy, and care for the environment. Seasonal variety was mentioned as a contextual factor related to local food [12]. Consumers, for example, in general perceive local food as healthier, more nutritious, and generally of higher quality [18,22,23]. Their preferences for local food are furthermore often positively related to consumer ethnocentrism [24], whereas consumers’ price consciousness often poses a barrier for local food consumption [17]. While local and seasonal food is frequently associated with environmental benefits, resulting in the of use environmental concern as a common motive [1,7,25,26], respective findings regarding the relevance of environmental motives as drivers in the literature are often ambiguous [27].

Although the extant literature assesses the relevance of single motives or groups of related motives in parallel efforts, it falls short of integrating the variety of relevant motives to identify each motive’s relative importance as a driver for sustainable food choices that combine local and seasonal attributes. Combining different motives can however potentially reveal trade-offs between them [13]. To our knowledge, studies evaluating the motives underlying the valuation of seasonality in combination with aspects related to origin are scarce. Most research focuses on specific and singular sustainability-related food options [28]. The few studies which combine different sustainability-related aspects (i.e., local and organic production) assessed consumer preferences for the different options by means of a choice-based conjoint (CBC) analysis [5,29]. While limited, the number of studies that investigate the choice of in-season food exclusively or in combination with organic or local food choices do demonstrate the relevance of considering this combination of attributes for a sustainable food choice [1,30,31]. As such, the study of Foster et al. [31] claims that a strong focus on seasonality exclusively is unlikely to deliver large environmental benefits.

Researchers such as Lazzarini et al. [32] and Aldaya et al. [33], among others, emphasize that a focus on local food alone is insufficient to reduce environmental impacts. Consequentially, to reduce environmental impacts regarding the primary energy use (PEU) [30] and water use [31], it is relevant to consider both locality and seasonality in the food choice [28,33]. With this study, we thus want to bridge this research gap and identify which motives are relevant to drive a consumer’s preference for and choices of food that is local compared to food that is both local and in-season (and thus more environmentally sustainable). More specifically, we want to empirically investigate whether environmental motives, for example, compared to socioeconomic motives, become (more) relevant for local food options that offer additional environmental benefits by also being produced in season. Methodologically, in line with previous research on local organic food choices [5,29], we presented the different food options to consumers within a CBC experiment. We thereby aim to address divergent findings in the literature regarding the relevance of environmental motives to drive local food choices [17,34,35]. 

Thus, we first investigate (a) the concepts of seasonal and local food and (b) the relationships of motives and barriers and food choices in the context of local seasonal food and local non-seasonal food to develop testable hypotheses for the empirical study. We use a CBC analysis to measure an individual’s preference for local seasonal food in combination with the assessment of five motives and one barrier for choosing these foods. We then analyze the choices of individuals and the relative influence of the different motives. Accordingly, we analyze the relevant relationships by integrating the motives as independent variables and the choice of local seasonal food as a dependent variable in a structural equation model. Based on the insights gained on the relevance of different motives to drive consumer preferences for and choices of local and seasonal food options, we followingly aim to derive effective and evidence-based recommendations to assist policymakers and marketers in the communication and target-group-oriented promotion of sustainable food consumption.

## 2. Conceptual Model

### 2.1. Concept of Local and Seasonal Food

The definition of “local” in the context of food varies across studies, ranging from references to travel distances, political boundaries, and specific criteria to more holistic approaches related to, e.g., ethical dimensions such as personal relations [12]. Accordingly, there is no consensus on the definition of what is local [36]. The extant literature reveals that, in some cases, researchers avoid defining the term for consumers and instead instruct study participants to respond to questions according to their perception of what is local [36], or examine their perceptions by providing different definitions of local food, i.e., produced within a certain distance, within a state or a country [5,21]. In other cases, while acknowledging definitions of local food according to travel distances, researchers often use the domestic origin of food as a proxy for its locality [18,32,37], and domestic food as an example for local food [1,33]. This rather broad understanding of local as domestic presents an important driver for the demand for local food [37]. While consumers have a low country of origin accuracy across many product categories [38], they tend to use domestic origin cues as a heuristic to evaluate unprocessed food in terms of healthiness, quality, or environmental footprints [22,32]. This especially holds true for developed countries, where consumers tend to prefer domestic products [29]. Considering the domestic origin as a salient option for defining, demanding, and evaluating local food, we thus define, for this study, local food according to the political boundaries of a country and use domestic and local food synonymously.

With regards to seasonal food, there are different perceptions of what is seasonal, resulting in a production-oriented global definition, and a consumer-oriented local definition [27,31]. The global definition is production-oriented and views seasonal food as food that is outdoor-grown or produced during the natural growing period for the country where it is produced [27,31]. This definition applies to seasonal foods produced either domestically or overseas. In contrast, the local definition links a local production to local consumption, thus defining seasonal food as produced and consumed in the same climatic zone without high energy use for climate modification such as cold storage and heated glasshouses [27,31]. For our study, we rely on the consumer-oriented local definition of seasonal, as it considers the energy use for climate modification and thus encompasses a perspective that is more likely to deliver environmental benefits, according to Brooks et al. [27].

The overall environmental performance of local seasonal food depends upon the selection of indicators under research (i.e., PEU, footprints of water, land, material use, and carbon, as well as emission intensity) [5]. As such, the study of Canals et al. [30] found that, in the case of apples, there is little difference in the PEU of a seasonal imported apple and a non-seasonal domestic apple due to storage loss. Furthermore, Brooks et al. [27] highlight that low production standards of a product produced in season can result in higher environmental impacts compared to state-of-the-art non-seasonal production. This finding coincides with the claim that the emission intensity of production dominates the change in transportation emissions following a policy intervention related to food miles, e.g., in the context of vegetable oils [39]. Furthermore, the environmental costs must be assessed case-by-case [27] and require multi-product approaches to identify benefits available from a general shift to seasonal food [31]. Product-specific examples do often include apples and tomatoes. Apples were, as mentioned earlier, used as a case product to compare the PEU related to the transport of imported seasonal and storage of domestic non-seasonal food consumption [30]. They were further used as an exemplary product to analyze consumer preferences for organically and locally produced apples using a CBC analysis [21] and compare biases [22], as well as the perceived environmental impact [32] for domestic versus imported apples. Amongst others, tomatoes have been investigated with regard to their field production [40] as well as with regard to the import of Moroccan tomatoes compared to non-seasonal French tomatoes [41]. 

### 2.2. Consumer Preferences and Food Choice

There are numerous examples from the literature analyzing the influence of motives or perceptions underlying consumer preferences [17,20,26,36,42]. These studies are usually based on theoretical frameworks such as value theory [43], theory of reasoned action (TRA) [44] as well as theory of planned behavior (TPB) [45], and alphabet theory that combines the value-belief-norm (VBN) theory [46] and the attitude-behavior-context (ABC) theory [47]. In accordance with the underlying theoretical frameworks, beliefs, norms, and attitudes are often taken as a proxy for the perceptions of consumers. Attitudes towards local food are oftentimes used as a proxy for a consumer’s preference [23,42]. Other studies use behavioral variables such as the intention and willingness to buy local foods [26] or self-reports of past behavior [17] as a proxy for an individual’s food choice. The CBC analysis presents an alternative to these scale-based measures of preferences and has been recently used to estimate preferences in food choices that contain different product attributes, such as organic and country-origin cues [29] and locality labels [5]. 

### 2.3. Motives

International research shows that the motives underlying local and seasonal food choices are driven by values, beliefs, and attitudes related to socioeconomic, health, and environmental aspects. Numerous studies did for example reveal the belief of supporting the local economy and farmers by opting for local food [48,49,50] as relevant motive. Biases in the perception of the product quality, which can be explained by the domestic country bias [50], lionization [36], and halo effects of local food [51], further play a role for local food choices. Individuals can also choose seasonal and local food to preserve local heritage and tradition [52] or because they desire authenticity [18]. Regarding the role of environmental concerns, Tobler et al. [1], for example, concluded that numerous reasons are underlying ecological consumption behavior, of which not necessarily all have to focus on the environmental outcome of the behavior [1]. Correspondingly, Brooks et al. [27] found that reducing the personal environmental impact often plays a secondary, if not tertiary, role for the purchase. We followingly elaborate on the different motives in detail to develop our research hypotheses.

#### 2.3.1. Consumer Ethnocentrism

Consumer ethnocentrism is defined as, “*beliefs held by consumers about the appropriateness, indeed morality, of purchasing foreign-made products*” [24]. The construct is based on the formation of “we-group” feelings, which define the in-group as the focal point, and all out-groups are judged in relation to it [53]. The construct aims to reflect normative beliefs concerning the appropriateness of buying domestic products compared to the inappropriateness of buying foreign products [54]. Accordingly, ethnocentric consumers are inclined to view the purchase of imported products as wrong, as according to them it affects the domestic economy and is not in congruency with in-group feelings of belongingness to the own society and patriotism [24]. A consumer’s ethnocentrism gives the individual a sense of identity, a feeling of belongingness, and an understanding of what purchase behavior is unacceptable or acceptable in the in-group [24].

Consequentially, the construct presents a key factor influencing the preference of consumers for domestic over imported products [50]. The perception of supporting the local economy is one of the most common drivers investigated in the context of local and seasonal food consumption [23,29,42,49]. Empirical results show, for example, a positive relationship between consumer ethnocentrism and the attitude towards local food consumption [42]. Furthermore, Fernández-Ferrín et al. [49] found that ethnocentrism influences the valuation of local–traditional–regional food products. Studies investigating the role of ethnocentrism for the choice of not only domestic but also seasonal products are scarce, despite the conceptual relation of seasonal and local food. The predictive power of the concept for the choice of seasonal food is yet to be studied.

Based on the conceptual definitions and empirical evidence for the influence of ethnocentrism on the choice of domestic food, we assume that consumers’ ethnocentrism also drives choices of a local seasonal food choice. Details on measurements are available in Section 3.3. Motive Measures and in Appendix B, Table A1.

**Hypothesis** **1** **(H1).**
*Consumers’ ethnocentrism positively influences consumers’ preferences for and choice of food that is local and in season.*


#### 2.3.2. Green Consumer Values

Green consumer values are defined as the “*tendency to express the value of environmental protection through one’s purchases and consumption behaviors*” [55]. The concept is based on the motivated reasoning process that consumers with stronger green consumption values prefer environmentally friendly products. In the conceptual development of green consumer values, Haws et al. [55] refer to the Theory of Basic Values [43,56,57] and the Self-perception Theory [58]. Accordingly, individuals with green consumer values use them as guiding principles for the purchase of environmentally friendly compared to traditional products [55]. 

Concepts addressing the environmental concerns and consciousness of individuals (i.e., ethical sustainability) are often used in the context of local food consumption, as consumers associate local food with shorter transport distances and reduced GHG emissions [1,7,25,26]. In the context of seasonal food, Tobler et al. [1] found environmental motives underlying the participants’ willingness to eat seasonal fruits and vegetables. These motives are partly covered by evidence from life-cycle analysis (LCA) on seasonal and local food reporting improvements in the performance of single environmental indicators [30,31,39,41]. A recent study further showed that from all consumer segments based on knowledge about sustainable food consumption, the segment focusing on origin attached the highest relevance to origin, transportation, and seasonality [59]. There are, however, also a number of studies that found no empirical evidence for the influence of environmentally driven motives in the choice of local products [17,27,34,35].

Consequently, studies reveal divergent results regarding the influence of environmental motives, such as green consumer value, on the choice of local products. We assume that adding seasonality as a product attribute to local food choices can add to the perceived environmental benefits of the food choice and enhance their green consumer value. Thus, we hypothesize that green consumer values will have a positive influence on the choice of local food that is also seasonal.

**Hypothesis** **2** **(H2).**
*Green consumer values positively influence consumers’ preferences for and choice of food that is local and in season.*


#### 2.3.3. Local Identity

“*A local identity consists of mental representations in which consumers have faith in and respect for local traditions and customs, are interested in local events, and recognize the uniqueness of local communities; broadly, being local means identifying with people in one’s local community*” [60]. The concept is based on the optimal distinctiveness theory, suggesting that the diagnosticity of a primed identity can be implicitly affected by whether people engage in integrative processing compared to differentiative processing [61]. In the context of product choices, Zhang and Khare [60] propose that a more accessible local identity influences their preferences for local products. 

In line with this proposal, studies found that local identity predicts the valuation and purchase of local food products [62,63]. Local food is perceived as a way to preserve local heritage and traditions according to Seyfang [52]. Thus, we assume that based on the conceptual definition of identity, individuals who, for example, care for local traditions and customs are more likely to choose local food over imported food. This is due to the ambition of these individuals to integrate, e.g., their food choices with the local identity’s characteristics [60].

Local identity and its effects on not only local but also seasonal food consumption has received scant research attention. As local food can address a consumer’s local identity, we, however, assume that seasonal food can be specifically linked to the season the food is typically harvested (e.g., apples in autumn), as well as to regional and cultural history, e.g., Wachau apricots [64]. Based on the conceptual overlaps of local and seasonal food, we believe that the effect of local identity predicting the valuation and purchase of local food is even more pronounced for local seasonal food. Hence, we hypothesize the following effect:

**Hypothesis** **3** **(H3).**
*Consumers’ local identity positively influences consumers’ preferences for and choice of food that is local and in season.*


#### 2.3.4. Authenticity

Authenticity in the context of products can be conceptualized as perceived brand authenticity, defined as, “*the extent to which consumers perceive a brand to be faithful towards itself (continuity), true to its consumers (credibility), motivated by caring and responsibility (integrity), and able to support consumers in being true to themselves (symbolism)*” [65]. As seen in the conceptual definition, authenticity can relate to the self, e.g., by being true to one’s self, or external entities, e.g., by projecting one’s beliefs, expectations, and perspectives onto an entity [66]. In order to create a narrative sense of the self, consumers follow practices, such as authenticating acts or authoritative performance [67]. These practices include creating agency through purchases or creating and sustaining shared traditions [67]. Accordingly, consumers seek authenticity in consumption acts [67,68]. Hence, Morhart et al. [65] concluded that consumers will respond positively to brands that they perceived as authentic. Riefler [69] further found that the positive effect of authenticity can even mitigate the competitive disadvantages of global brands in localized markets. This indicates the relevance of authenticity as a key determinant for food choices.

In the context of local and seasonal food, consumers also pursue social and locational authenticity through consumption patterns [70,71]. Indeed, the study of Bryła [72] showed that the perceived authenticity of a product is strongly connected to its origin and sale in the region of origin. In the context of brands, localness was found to be an important brand attribute that helps to drive authenticity perceptions [73]. Furthermore, Ditlevsen et al. [18] found the desire for authenticity to be an important motivation for consumers of local foods.

Stemming from the conceptual nature of authenticity and the use of the concept in the local food context, we believe that authenticity can likewise be a motive for the choice of food combining local and seasonal attributes. This assumption is based on the connection of seasonal food to locational authenticity, as in certain harvesting seasons the consumption of certain seasonal food can be linked to traditions and local heritage. As the authenticity of a local and seasonal product is based not only on the product itself but on agricultural aspects, we propose that the perceived authenticity of local agriculture and local seasonal products influences food choice.

**Hypothesis** **4** **(H4).**
*Consumers’ perceived authenticity of the local agriculture and local food positively influences consumers’ preferences for and choice of food that is local and in season.*


#### 2.3.5. Healthiness Bias

The domestic country bias [50] includes the perception that local food is healthier, which is thus also referred to as healthiness bias [22] and more recently as lionization [74]. It can be defined as a “*systematic tendency to evaluate domestic products as healthier than equivalent foreign products*” [22] or as “*a belief that local foods possess superior taste and quality*” [36] within the food context. The construct, like ethnocentrism, is conceptually based on the formation of an in-group and an out-group being judged using the in-group as a reference [24,50]. The healthiness bias can be manifested in a consumer’s perception and purchase intention of products [75]. It presents a self-beneficial motivation for the food choice [36]. As Balabanis and Diamantopoulos [50] found, the bias can be production/origin-specific. 

So far, studies have mainly focused on origin-specific biases, which assess whether consumers perceive food products differently if they are domestic [18,22,23]. Research on the production-specific bias (for example, in the context of seasonal food) is rather limited and focuses on organic food. A review paper on that can be found in Aertsens et al. [76]. Regarding seasonal food, Tobler et al. [1] found their study participants to be convinced that seasonal fruits and vegetables taste better. Gineikiene et al. [22], among others, provided empirical evidence for the relation of the healthiness bias and the choice for domestic products. They further found that the positive effect of the healthiness bias on the choice of domestic products holds through different product categories such as apples, tomatoes, bread, and yogurt [22]. Furthermore, lionization as a part of the locavorism concept [74] predicts the attitude towards buying local food [36].

Hence, based on the conceptual nature of the healthiness bias or lionization and its relation to domestic food consumption, as well as the empirical evidence for the predictive quality of the belief on domestic food choice, we develop the following hypothesis, again considering the conceptual relation of seasonal and local food:

**Hypothesis** **5** **(H5).**
*Consumers’ perception of domestic and seasonal food as more natural, healthier, and tastier positively influences consumers’ preferences for and choice of food that is local and in season.*


#### 2.3.6. Price Consciousness

Price generally is one of the most important marketplace cues due to its presence in all purchase situations [77]. Price consciousness is defined as *“the degree to which the consumer focuses exclusively on paying a low price”* [77] and represents one of several price-related constructs. The construct stems from the marketing literature and is one of the constructs consistent with a perception of price in its “negative role” as an outlay of economic resources [77]. Consequentially, a high price can function as a barrier to the purchase of a product and thus presents an important food choice motive [78]. In a segmentation study, Scalvedi and Saba [7] found that non-local consumers were motivated in their food choice mainly by the brand and price. Accordingly, several studies associate price as a barrier to local food purchasing [17,26,34]. 

Based on the findings from substantive literature, we believe that the price of a food product can outweigh the utility of less present cues, such as origin and seasonality. Therefore, we assume, the price presents a barrier to the local seasonal food choice, leading to the following hypothesis:

**Hypothesis** **6** **(H6).**
*Consumers’ price consciousness negatively influences consumers’ preferences for and choice of food that is local and in season.*


### 2.4. Conceptual Model

The conceptual model for this study, built upon the developed hypotheses following the literature review, is depicted in Figure 1. It assumes that the consumer preferences and choice of local and in-season food depend on a consumer’s ethnocentrism, green consumer value, local identity, perceived authenticity of the local agriculture, healthiness bias and price consciousness. These motives are then also used in a conceptual model, which includes the preference of consumers for local but non-seasonal food as a dependent variable. The dependent variable of the conceptual model “Local and (non)local seasonal food choice” (LC) is measured by means of part-worth utilities; the latter are approximated by the CBC analysis (see Section 3.2).

In addition to the motives, for which we developed a hypothesis regarding their influence on local and (non)seasonal food choice, we included two control variables to balance the relation between motives driving and hampering the respective food choice. These are a consumer’s globalization attitude, i.e., a positive evaluation of economic globalization [79], and global identity, i.e., a mental representation in which consumers see themselves as part of a global community [80]. These constructs were measured according to Spears et al. [79] (globalization attitude, three items) and Makri et al. [81] (global identity, four items) (see Appendix B).

## 3. Materials and Methods

### 3.1. Data Collection and Sample

The data to empirically investigate consumer motives and preferences were collected by an external panel provider in November 2019, using an online survey with an embedded discrete choice experiment with a representative sample of 499 Austrian households (quota sampling). Due to their harvesting season, we decided to conduct data collection in November to include a seasonal (apples) and a non-seasonal product (tomatoes) into our experimental design. Austria as an exemplary country imports apples and tomatoes from a number of countries, despite high self-sufficiency rates of these products [82]. To ensure representativeness and variance in the sociodemographic profile of respondents, quotas for age (range 18–65), gender, education, and residence were set accordingly. The sample was further screened for the (at least partial) responsibility of respondents for their household’s grocery shopping and the consumption of the case product. Before the launch of the actual survey, about 10 individuals were asked to test the survey design and context for comprehensibility and functionality. 

The survey consisted of (i) an assessment of the sociodemographic characteristics of the respondents and (ii) a discrete choice experiment, which is a common method for the analysis of consumer preferences for different product attributes [83,84]. The design and analysis of the CBC to assess consumers’ preferences and the measurement of underlying motives are elaborated in detail in the following section. The final part (iii) of the questionnaire covered values, beliefs, and attitudes assumed to be underlying motives of choice responses using pre-developed scales from previous research. As a measure against the common method bias, the survey included a methodological separation of measurement in the study design (i.e., use of different scale formats to assess the independent variables (e.g., motives) and dependent variable (i.e., consumer preferences) [85,86].

### 3.2. Discrete Choice Experiment: Design and Analysis

Discrete choice experiments are based on the random utility theory [87,88,89]. Respondents are asked to choose out of a set of product options the most appropriate one (or none of them). The product options combine different product attributes sourced from a defined attribute set [90]. Based on the random utility theory it is assumed that the respondents will select the product option that represents the maximum utility perceived. Thus, the CBC analysis aims to reveal the weight of preferences consumers have towards single product attributes [91]. This method has the key advantage of further revealing apparent trade-offs made between the different product attributes and levels compared to an assessment of consumer preferences that uses hypothetical questions. These experiments are, thus, less influenced by response styles from scale use [92] and the social desirability bias [93]. Consequently, discrete choice experiments are a frequently used method within consumer research, with several application examples within the context of local food consumption [5,29]. 

Accordingly, this study also employed a discrete choice experiment, analyzing data by means of CBC analysis, to (1) realistically simulate choice sets (including a non-choice option) of grocery shopping, (2) estimate the importance of individual product attributes, and (3) estimate part-worth utilities of individual attribute values using Hierarchical Bayes estimation for each respondent. The first step included an assortment survey of apples (*n* = 79) and tomatoes (*n* = 80) conducted in October 2019 in Austria’s retail sector within actual purchase settings. This survey included the assessment of product attributes such as origin, price, and packaging with relevant attribute levels (e.g., price range, indications country of origin, package weight) used for a realistic design of product options. The second step in the design process for the discrete choice experiment included the selection of relevant characteristics of apples, representing a seasonal fruit variety, and tomatoes, representing a non-seasonal vegetable variety, according to the assortment survey in step one. Both products have high market penetration and are available as regional and imported products. Furthermore, as mentioned earlier, apples and tomatoes are often used as case products in studies regarding consumer preferences and environmental impacts of different food choices [22,41]. The product options of the CBC analysis were combined from four product attributes, including their single attribute levels. This resulted in a total of 90 (3 countries of origin × 5 price points × 3 packaging weights × 2 types of production (organic/conventional)) product options for apples and tomatoes (see Table 1).

These product options were randomly selected and bundled into 10 choice sets designed following an online grocery store. A choice set provided the respondents with three product options and a fourth non-choice option, which allowed respondents to refuse the hypothetical purchase. Each respondent was randomly assigned to one of two choice experiments: one including seasonal food choices (apples) and one including non-seasonal food choices (tomatoes) by the time of the assessment in November 2019. The part-worth utility that respondents attributed to domestic (Austrian) apples and tomatoes functioned as an independent variable representing the choice of local seasonal food and local non-seasonal food. All steps in the process including random selection of choice sets, random assignment of respondents, and approximation of part-worth utilities were executed by means of the choice analytics survey software “Sawtooth Lighthouse Studio 9.7.2”.

### 3.3. Motive Measures

The six motives functioning as independent variables were assessed using pre-developed scales selected based on previous research findings regarding their suitability and validity within the context of local and seasonal food choices. 

As mentioned above, consumers’ ethnocentrism was measured by four items of a short version [94] of the CETSCALE [24]. The environmental motive was assessed using the GREEN scale [55], which measures green consumer value with six items. As respondents are often prone to provide socially desirable answers regarding their environmental responsibility, we reversed the wording of two items and deleted the item “I would describe myself as environmentally responsible”. The local identity of respondents was measured using four items as applied by Makri et al. [81] and previously developed by Tu et al. [80]. The perceived authenticity of the domestic agriculture and domestic and seasonal food was measured by four items adapted from Morhart et al.’s [65] brand authenticity scale. To measure the healthiness bias and domestic country bias, six items were used combined from the measures of Gineikiene et al. [22] and Aprile et al. [23]. To assess price consciousness as a possible barrier, two items were adapted from the measure of Koschate-Fischer et al. [95]. All motives were assessed on 7-point Likert scales ranging from 1 = strongly disagree to 7 = strongly agree (items shown in Table A1 in Appendix B). 

### 3.4. Statistical Methods

The preference of consumers for domestic origin as a product attribute in both the seasonal and non-seasonal choice experiment was determined by the part-worth utility for this product attribute. This part-worth utility each individual attributed to Austria as the country of origin effect was assessed with the application of a Hierarchical Bayes estimation. A descriptive analysis of the motives and an analysis of their correlations were conducted using SPSS 26. The relationships between the motives and the choice preference for domestic and seasonal or non-seasonal products as specified in the conceptual model (Figure 1) were assessed using the Maximum Likelihood Estimation (MLE) with structural equation modeling in LISREL version 9.30. We chose a covariance-based SEM approach, as we primarily focused on the empirical confirmation of the respective relationships and their relative importance indicated by the relevant path coefficients, see [96,97,98]. 

## 4. Results

### 4.1. Profile of the Respondents

The sociodemographic characteristics of the representative sample of the study in November 2019 are described in Table 2. The sample was also grouped according to the case product to which the respondents were assigned in the CBC analysis. It is largely representative of the Austrian population, and deviations in view of age and gender were negligible. In view of the degree of urbanization, the rural population was underrepresented; concerning household size, one-person households were underrepresented as well, and the same was true for education in view of university degree. Overall, the sample consisted of 52.7% female and 47.3% male participants with an average age of 46.6 years. With regards to educational attainment, most participants absolved an apprenticeship (40.9%). Thirty-one percent of the sample completed high school or higher education. The residential areas of the respondents were evenly distributed between urban, suburban, and rural areas. Most of the respondents lived in two-person households (40.5%). Including these variables into the structural equation model (see Section 4.5) provided non-significant outcomes for age, gender, household size, etc. Accordingly, sociodemographic variables had no influence on the results of our causal model. Although there were some deviations in our sample in comparison to the total population, these deviations did not affect the reliability of our results.

### 4.2. Results of the CBC Analysis

Hierarchical Bayes estimation was used to approximate the individual preferences of respondents regarding four product attributes: country of origin, production type, price, and package size. The part-worth utilities were estimated for each product attribute and the relevant attribute levels (see Table 3). The results show that origin had the highest part-worth utility in both choice experiments (48% for apples, 39% for tomatoes), followed by the price, packaging weight, and type of production. The part-worth utilities for all four product attributes sum up to 1. The lowest part-worth utility was set to zero within each attribute. Consequently, as seen in Table 3, respondents credited less utility to imported apples and tomatoes that are conventionally produced, have a high price and medium packaging weight. Accordingly, the higher the resulting part-worth utility is, the greater the benefit the consumer perceived for him or herself provided through the specific product attribute. Consequentially, these greater benefits result in a greater likelihood that the consumer will purchase a product with the relevant characteristics (attribute level) he or she perceives as beneficial [99]. Mean, standard deviation, minimum, and maximum can be taken from Table 3; it shows the wide range of values based on Hierarchical Bayes estimation. According to Hypotheses 1–6, we assume that the importance of the attribute “Local” (i.e., domestic origin of the product) is influenced by the motivational structure of respondents (research model in Figure 1).

A detailed analysis of the single choices revealed that 67% of the chosen apples and 63% of the chosen tomatoes were of Austrian, thus local, origin. If a choice set did not offer any domestic choice options, respondents refused the hypothetical choice in 43% of the cases in both discrete choice experiments.

### 4.3. Descriptive Analysis of Motives

Table 4 lists the results obtained from the descriptive analysis of the motives assessed (sample size *n* = 499). The mean value of all six latent variables ranged from 4.15 to 5.49, and the standard deviation ranged from 1.08 to 1.43 on a 7-point Likert scale. The mean values of all the variables were above the midpoint of 3.50. Authenticity scored the highest with a mean of 5.49, compared to price consciousness that scored the lowest with a mean of 4.15. The dispersion values, reported through the standard deviation, were the highest for price consciousness and the lowest for green consumer value. 

### 4.4. Assessment of the Measurement Model

Validity and reliability were determined as part of the measurement model assessment [100]. This includes an assessment of the constructs’ convergent and discriminant validity by computing composite reliabilities (CR) and average variance extracted [101]. All CR scores exceeded the recommended threshold of 0.7, suggesting that the constructs had good internal consistency in both samples (Table 5). The AVE values, except for the construct of green consumer value, were all above the threshold of 0.5. AVE values below 0.5 indicate that the measurement error accounts for a greater amount of the variance occurring in the indicators than the variance in the latent variable account for [100]. In the case of the green consumer value construct, the AVE score below 0.5 was related to the low magnitude of the loading λ of the items env3 and env5. After removing these items from the measurement, the AVE of green consumer value exceeded the critical threshold of 0.5 (AVE_apples_ = 0.635, AVE_tomatoes_ = 0.641). Thus, both measurement models indicated a reasonable convergent validity.

Table 6 shows the results of the discriminant validity assessment following the Fornell and Larcker [101] criterion of comparing the correlation between constructs and the square root of the AVE (along the diagonal) of that construct. The square root of the AVE for each construct was greater than the correlations, indicating that each construct had discriminant validity in both samples [101].

The model fit indices of the measurement models were tested (Apple sample: chi-square = 774.544, 399 df; CMIN/df = 1.94; TFI = 0.904; RMSEA = 0.061; CFI = 0.917; Tomato sample: chi-square = 749.906, 399 df; CMIN/df = 1.88; TFI = 0.909; RMSEA = 0.059; CFI = 0.922), yielding in an acceptable fit considering commonly used thresholds [102].

### 4.5. Estimation of the Structural Model

After the assessment of the measurement model, the structural models for both seasonal and non-seasonal food choices were estimated, using summated scores (factor scores) of the six independent variables and fixing error variance at a level appropriate to its coefficient alpha reliability, i.e., 1 − α [103]. We obtained the factor scores by performing an exploratory factor analysis (EFA) in SPSS 26, extracting the data based on the principal axis factoring method [104] with varimax rotation and Kaiser normalization (KMO = 0.905; df = 435; *p* = 0.001) [105] (see Appendix A for factor loadings). The total variance explained was 63.79%, indicating a well-explained factor structure. Given that there was no single factor, and the first factor did not represent the majority of the variance, we can assume that the relationship between the variables was not inflated by the common method bias (CMB) [86,106,107]. According to Hypotheses 1–6, we assume that the dependent variable LC of the model (i.e., the part-worth utility for the attribute “Domestic”) should be influenced by the motives and the price barrier. We, therefore, analyzed the hypothesized directions and strength of relationships captured by the standardized coefficients γ (gamma). Table 7 summarizes the results of the structural model analysis for the hypothesis testing. The data show that consumer ethnocentrism and the healthiness bias were significantly and positively related to the preference for and choice of both seasonal and non-seasonal domestic food. Followingly, Hypotheses H1 and H5 are supported. When comparing the strength of the relationships captured by the standard coefficient γ, Table 7 indicates that the influence of consumers’ ethnocentrism was positive and stronger for non-seasonal food choices (γ_tomatoes_ = 0.383) compared to seasonal food choices (γ_apples_ = 0.268), while the influence of the healthiness bias was stronger for seasonal food choices (γ_apples_ = 0.193 vs. γ_tomatoes_ = 0.182). The results further show that the preference for and choice of seasonal food was slightly influenced by the perceived authenticity of the local agriculture (γ_apples_ = 0.130), but not by the local identity. In contrast, the local identity had a low, positive, and significant influence on the preference for and choice of non-seasonal food (γ_tomatoes_ = 0.165). Thus, in this case, H3 is supported for the model including local seasonal food choice as the dependent variable, and H4 is supported for the model including local non-seasonal food choice as the dependent variable. The influence of price consciousness as a barrier on choice was supported for both seasonal (γ_apples_ = −0.306) and non-seasonal food (γ_tomatoes_ = −0.429). Accordingly, the negative impact of price seemed to be even stronger for non-seasonal food compared to seasonal food. This finding confirms H6, as the influence is significant and negative between the constructs. 

However, the explanatory power of these models was low, in particular for seasonal food choice where the motives and barrier explained only 25.8% of the variance in seasonal food choice. In addition, for non-seasonal food choice, the explanatory power of the model was rather low (45.8% of the variance). The control variables globalization attitude and global identity had no influence of both types of food consumption; γ was not significant. These variables were eliminated from the model. As mentioned above, the same can be said for the sociodemographic variables. These, too, did not influence LC.

## 5. Discussion

The present study aimed to identify the relative importance of different motives underlying local and seasonal food choices compared to non-seasonal food choices. It aims to derive effective and evidence-based recommendations for promoting environmentally friendly food choices. We assessed the motives using established scales from the literature and derived the consumer preferences for local and seasonal food (i.e., apples) as well as local but non-seasonal food (i.e., tomatoes) from the part-worth utility attributed to these choice options approximated by means of a CBC analysis (including Hierarchical Bayes estimation of individual part-worth utilities). The relationships between motives and preferences were then analyzed using structural equation modeling (see Figure 2a,b). The focal objective was to identify whether environmental motives drive choices of food that delivers potential environmental benefits, as the literature currently provides divergent findings regarding the relevance of such motives. By adding the seasonality aspect in the local food discourse, this study further addresses calls from studies emphasizing that local food choices alone are insufficient to ensure low environmental impacts of the consumption, as local food is only environmentally friendly when harvested in season and derived from sustainable production systems [32].

Our findings showed that, despite respondents having a strong tendency to express values of environmental protection through their purchase, these green consumer values did not influence their choice of local and seasonal (Figure 2a) nor local and non-seasonal foods (Figure 2b). This is in line with Tobler et al. [1], who found that environmental motives for the consumption of seasonal food did not have a significant influence on the transition from considering changing to actually adopting such consumption patterns. We believe that a reason for the lack of relevance of environmental motives could lie in the complexity of understanding which and how environmental benefits result from a seasonal food choice. As Tukker et al. [8] conclude, the assessment of the environmental impact gets more complicated when comparing local fruits and vegetables produced in energy-intensive greenhouses with the “food miles” that are accrued by alternatives grown on the field in distant locations. As a consequence, one has to consider not only the carbon but also the land, material, and water footprint for a holistic evaluation of possible impact reductions related to seasonality. The complexity of this evaluation could hamper the consideration of the seasonality aspect in general. Literature findings show that consumers perceive the consumption of seasonal fruits and vegetables as less relevant to the environment than, for example, excessive packaging and more relevant than the purchase of organic food, which is in contrast to LCA results [1,108]. This might be related to an underestimation of the environmental impacts of out-of-season production [32]. Consumers seemingly attribute more relevance to the environmental impacts resulting from food transport and regard local food as more environmentally friendly due to short transportation distances [25]. 

While our results showed that environmental motives influence neither the choice of seasonal or non-seasonal local food, they indicate that authenticity plays a more relevant role when choosing seasonal local food, while local identity is more relevant when choosing non-seasonal local food. We believe that this difference stems from the conceptual nature of both motives. Authenticity is a broad concept that is linked to not only the geographical origin of a product but also to traditions related to its production and marketing [63]. This layer of the authenticity motive, capturing aspects in addition to those regarding the origin of a product, could be the reason for the relevance of this motive for a seasonal food choice. The origin might not be the main aspect for the choice of food that is also seasonal, as some consumers might understand the local origin as a precondition for seasonality, according to the consumer definition of the concept [27,31]. The opposite might be true for non-seasonal local food, for which consumers then attribute even higher importance to the fact that the food is not seasonal but of a local origin, which they can relate to as part of their local identity.

In addition to revealing the relevance of authenticity and local identity as motives for seasonal and non-seasonal, local food choice, our study further confirmed the healthiness bias as the second most relevant motive in the context of local food choices. The descriptive results (see Table 4) correspond to previous findings, which indicate that local food is also perceived as healthier, better in taste, and more natural and nutritious [18,22,23,50,74]. By integrating this motive in a structural equation model to estimate its influence on consumer preferences, we confirmed the relevance of this bias not only for local food [22] but also for seasonal food. 

Both structural equation models revealed that consumer ethnocentrism is a key driver for a local food choice. This is in line with previous research [23,29,42,49]. Our results show that, especially, those consumers who want to support local farmers and agriculture reach for both seasonal and non-seasonal local products. A comparison of both models showed that consumer ethnocentrism is even more relevant for food choices that are non-seasonal but of local origin. This indicates that consumers might consider it more important where a product is produced compared to how it is produced, as the support of the local economy as a key driver is more dependent upon the location than on the type of production (i.e., indoor- or outdoor-grown). The relevance attributed to origin is also seen in the results of the CBC analysis, which accordingly revealed origin as the most relevant product feature, with Austria as the domestic country having the highest part-worth utility. The part-worth utility of origin was slightly higher for apples as a seasonal product than for tomatoes as a non-seasonal product. Consumers are probably more flexible regarding the country of origin when the product is non-seasonal. An additional analysis of the choice sets, however, revealed that consumers went for the non-choice option in 43% of the cases that offered no local option. The identification of motives that underlie the non-choice of consumers when confronted with non-local food options thereby opens an avenue for future research. In this context, certain biases towards countries of origin play a relevant role and should be considered in further studies. As such, future research should also consider these biases when addressing the reasons underlying the reduced importance attributed to a local origin of non-seasonal products and the increased flexibility regarding the country of origin.

The main barrier to buying local and seasonal or non-seasonal food is price consciousness, which was more relevant than any other motive. The models of both samples showed that consumers who want to, or have to, buy cheap tend to purchase fewer local foods, which corresponds to findings from the literature [17,26,34]. A comparison of the models showed that the price consciousness is lower for seasonal (Figure 2a) compared to non-seasonal, local food (Figure 2b). This might indicate that consumers, in the case that food is both local and seasonal, attribute less importance to the price as an attribute, whereas the opposite is true for local but non-seasonal products. The results from the CBC analysis indeed show that the price as an attribute is less relevant for seasonal apples than for non-seasonal tomatoes. Both samples attributed the highest part-worth utility to a low-price level, whereas this was repeatedly less important for seasonal products.

The above discussion of the key motives and barriers shows the demand for future research to investigate further drivers of food choices that combine locality and seasonality. According to the variance explained by our model, there are further influencing factors to be considered. As such, future research could consider the relevance of environmental knowledge [109] for the choice of seasonal and local food, as our research showed that environmental values, such as green consumer values [55], are not linked to a respective environmentally friendly food choice. As a consequence, researchers could investigate whether a certain level of environmental knowledge is positively related to the preference for and choice of food that is not only local but also seasonal. As the assessment of the environmental impact of food choice gets increasingly complicated [8], and the impact of an out-of-season production might be underestimated [32], a high level of environmental knowledge could facilitate this understanding and followingly drive consumers to opt for an environmentally friendly option. As a further avenue, future research could assess amongst others to what degree a consumer’s connectedness to nature [110], environmental identity [111], or ecological identity [112] influences his or her preference for and choice of seasonal products. As according to a consumer-oriented local definition seasonal food is outdoor-grown or produced during the natural growing period [27,31], consumers that feel connected to nature, or as a part of nature, could be more aware of the seasonality of different foods, which might be a motive to also opt for seasonal food. These environmental motives should, again, be integrated into analyses that include additional motives specifically related to seasonal food choices, such as the importance attributed to the foods’ taste and freshness [1]. When engaging research on seasonal food consumption, it is further relevant to not only consider motives for the commission of seasonal food choices but the omission of non-seasonal food choices. From an environmental perspective, benefits can also stem from reducing food choices with a potentially negative environmental impact. An interesting driver in this context could be a consumer’s past environmentally motivated consumption reduction [113]. Altogether, the inclusion of some of these variables could help to increase the explanatory power of our research model significantly, which is rather low in particular for seasonal food choice.

## 6. Conclusions

With this study, we aimed at contributing to a holistic concept of local food that encompasses seasonality. To reach this goal, we used a methodological mix of CBC analysis and SEM. This approach allowed us to obtain valid and reliable results leading to the following evidence-based recommendations. We conclude that policymakers and marketers should link the consumption of local and seasonal food to the contribution to the domestic economy and support of local farmers. Regarding the role of price as a barrier, the main challenge for marketing local and seasonal origin as a product attribute particularly lies in strengthening the willingness to pay more as a result. The branding and labeling of food should reflect the intrinsic qualities that consumers are seeking [17]; thus, we recommend marketers to consider the perceptions and expectations consumers hold towards local and seasonal food. In the communication, marketers should thus emphasize aspects related to product quality and the authenticity of seasonal food. 

As our study could not identify the relevance of motives related to environmental sustainability but to economic sustainability (expressed by the support of local farmers through local and seasonal food consumption), we recommend policymakers to adopt a holistic concept of local food. Embracing a “local seasonal food” concept can, according to Vargas et al. [28], force methodological approaches that address additional layers of sustainability, further allowing more concrete results to foster sustainable consumption.

Despite the contributions, this study must be considered under the following limitations. First, seasonality is dependent on the product and season; thus, this study was limited to the choice of specific case products. To decrease certain biases towards products, we would recommend increasing the varieties of in-season and out-of-season products and further replicate this study in a different season, as different products will be seasonal. Second, in the case of this study, the primary objective for the experiment was a realistic simulation of the currently available offer in the retail stores in Austria; therefore, we did not focus on biases towards the chosen countries of origin. However, these possible biases towards the selected countries of origin could influence the respondent’s choice. Third, regarding the motives, it would be interesting to assess the role of not only environmental values but also environmental knowledge regarding the actual environmental impacts of specific food choices. And forth, the data were collected in one specific, highly developed food market (Austria). Other markets that are not comparable to the Austrian market could deliver different results. This could be an interesting field of future research. Hence, we recommend future research to further investigate consumers’ understanding of seasonal food and its environmental impact and to conduct these studies in other food markets. More specifically, researchers could elaborate on which conditions in retail stores facilitate and foster the choice of seasonal food. Furthermore, the joint effect of seasonality and origin cues on consumer perceptions could be further investigated, as it is currently often practiced with organic and local attributes [29]. The lack of influence of environmental motives for the choice of seasonal products could be investigated by including further barriers in the model that might explain the gap between present environmental values and the limited consideration of those when choosing products.

## Figures and Tables

**Figure 1 foods-10-02715-f001:**
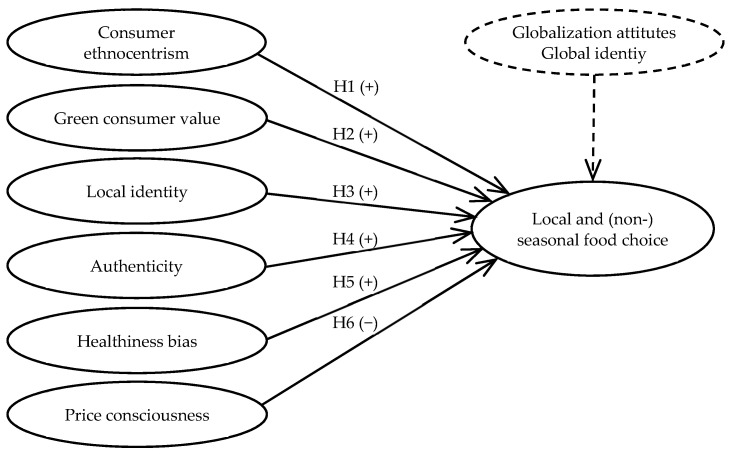
Conceptual research model.

**Figure 2 foods-10-02715-f002:**
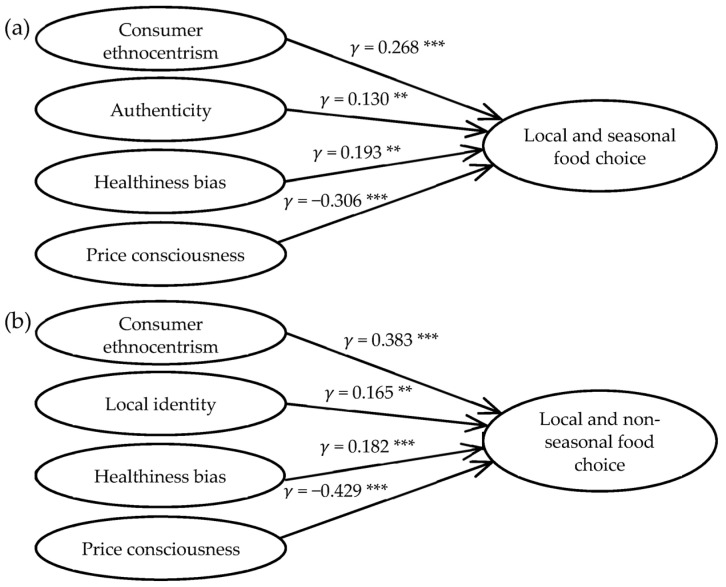
Evaluated research model including significant relationships between three motives and one barrier on (**a**) local and seasonal food choice and (**b**) local and non-seasonal food choice; ** *p* < 0.05, *** *p* < 0.001.

**Table 1 foods-10-02715-t001:** Design of the choice sets.

Attribute	Levels	Apples (*n* = 250)	Tomatoes (*n* = 249)
Origin	Domestic	Austria	Austria
	Country 1	Italy	Spain
	Country 2	Poland	Netherlands
Organic	no	conventional	conventional
	yes	organic	organic
Price	low	EUR 1.29	EUR 1.49
		EUR 1.79	EUR 1.99
		EUR 2.39	EUR 2.49
		EUR 2.89	EUR 2.99
	high	EUR 3.49	EUR 3.59
Package size	small	loose/singe	loose
	medium	Box of 6 pieces (750 g)	500 g
	large	1.5 kg bag or net	750 g

**Table 2 foods-10-02715-t002:** Sociodemographic characteristics.

		Apples (Seasonal)		Tomatoes (Non-Seasonal)		Total		Austria *
Variable	Sample Size	250		249		499		
	Description	Frequency	Percentage	Frequency	Percentage	Frequency	Percentage	Percentage
Gender	Female	129	51.6%	134	53.8%	263	52.7%	50.8%
	Male	121	48.4%	115	46.2%	236	47.3%	49.2%
Age (in years)	15–29	47	18.8%	46	18.5%	93	18.6%	15.3%
	30–44	64	25.6%	65	26.1%	129	25.9%	28.6%
	45–59	82	32.8%	81	32.5%	163	32.7%	32.4%
	60–75	57	22.8%	57	22.9%	114	22.8%	23.8%
Highest degree of education	Compulsory school	12	4.8%	19	7.6%	31	6.2%	17.6%
	Apprenticeship	106	42.4%	98	39.4%	204	40.9%	33.4%
	Vocational School	51	20.4%	58	23.3%	109	21.8%	14.4%
	Secondary school	49	19.6%	43	17.3%	92	18.4%	16.0%
	University degree	32	12.8%	31	12.4%	63	12.6%	18.6%
Degree of urbanization	Cities	87	34.8%	88	35.3%	175	35.1%	32.2%
	Suburbs	92	36.8%	82	32.9%	174	34.9%	27.7%
	Rural area	71	28.4%	79	31.7%	150	30.1%	40.1%
Household size	1	42	16.8%	52	20.9%	94	18.8%	37.8%
	2	108	43.2%	94	37.8%	202	40.5%	30.4%
	3	56	22.4%	51	20.5%	107	21.4%	14.6%
	4	29	11.6%	38	15.3%	67	13.4%	11.3%
	>5	15	6.0%	14	5.6%	29	5.8%	6.0%

* Source: Statistics Austria, https://www.statistik.at/web_en/statistics/index.html (accessed on 5 October 2021).

**Table 3 foods-10-02715-t003:** Estimated part-worth utilities (aggregated).

Attributes and Levels	Apples (Seasonal)				Tomatoes (Non-Seasonal)			
	Mean	Std. Dev.	Min	Max	Mean	Std. Dev.	Min	Max
Origin	0.474	0.211	0.032	0.864	0.393	0.203	0.012	0.790
Local	0.471	0.215	0.000	0.864	0.386	0.213	0.000	0.790
Country 1	0.157	0.073	0.000	0.381	0.029	0.040	0.000	0.242
Country 2	0.001	0.008	0.000	0.069	0.025	0.048	0.000	0.363
Price	0.235	0.158	0.021	0.728	0.326	0.181	0.026	0.818
low	0.235	0.158	0.021	0.728	0.326	0.181	0.026	0.818
medium-low	0.210	0.143	0.014	0.610	0.265	0.128	0.022	0.572
medium	0.142	0.102	0.005	0.505	0.202	0.088	0.016	0.414
medium-high	0.120	0.082	0.004	0.383	0.155	0.088	0.006	0.393
high	0.000	0.000	0.000	0.000	0.000	0.000	0.000	0.000
Package size	0.210	0.112	0.016	0.660	0.187	0.108	0.004	0.571
loose	0.177	0.111	0.000	0.660	0.173	0.115	0.000	0.571
small	0.018	0.059	0.000	0.462	0.012	0.057	0.000	0.502
large	0.152	0.107	0.000	0.452	0.103	0.062	0.000	0.380
Organic	0.080	0.085	0.000	0.597	0.094	0.088	0.001	0.545
no	0.007	0.027	0.000	0.234	0.013	0.032	0.000	0.275
yes	0.073	0.087	0.000	0.597	0.082	0.094	0.000	0.545

**Table 4 foods-10-02715-t004:** Description of focal motive constructs.

Construct	No. of Items	Apples (Seasonal)		Tomatoes (Non-Seasonal)	
		Mean	Std. Deviation	Mean	Std. Deviation
Consumer ethnocentrism	4	5.208	1.236	5.137	1.272
Green consumer value	5	5.183	1.125	5.060	1.153
Local identity	4	5.156	1.138	4.912	1.148
Authenticity	4	5.493	1.114	5.342	1.069
Healthiness bias	6	5.259	1.085	5.173	1.038
Price consciousness	2	4.152	1.427	4.257	1.512

**Table 5 foods-10-02715-t005:** Measurement parameters, construct reliability, and AVE scores.

Construct	Item	Apples (Seasonal)			Tomatoes (Non-Seasonal)		
		λ	CR	AVE	λ	CR	AVE
Consumer ethnocentrism	cet1	0.817	0.872	0.631	0.846	0.882	0.652
	cet2	0.754			0.767		
	cet3	0.830			0.854		
	cet4	0.774			0.759		
Green consumer value	gcv1	0.862	0.838	0.635	0.815	0.842	0.641
	gcv2	0.811			0.839		
	gcv4	0.709			0.745		
Local identity	lid1	0.771	0.861	0.608	0.813	0.846	0.580
	lid2	0.770			0.712		
	lid3	0.822			0.773		
	lid4	0.755			0.744		
Authenticity	auth1	0.800	0.915	0.730	0.851	0.905	0.705
	auth2	0.884			0.850		
	auth3	0.874			0.884		
	auth4	0.857			0.770		
Healthiness bias	hb1	0.864	0.914	0.703	0.738	0.896	0.633
	hb2	0.895			0.786		
	hb3	0.705			0.690		
	hb4	0.839			0.795		
	hb5	0.791			0.821		
	hb6	0.687			0.776		
Price Consciousness	pri1	0.669	0.813	0.692	0.761	0.797	0.663
	pri2	0.968			0.884		

Note: λ = factor loading, CR = Composite Reliability, AVE = Average Variance Extracted.

**Table 6 foods-10-02715-t006:** Discriminant validity of the measurement model.

	Apples (Seasonal)						Tomatoes (Non-Seasonal)					
	1	2	3	4	5	6	1	2	3	4	5	6
1 CET	0.794						0.808					
2 GCV	0.521	0.797					0.652	0.801				
3 LID	0.642	0.420	0.780				0.526	0.417	0.761			
4 AUTH	0.687	0.523	0.743	0.854			0.654	0.592	0.653	0.840		
5 HB	0.670	0.517	0.506	0.669	0.838		0.655	0.527	0.590	0.738	0.796	
6 PRI	−0.128	−0.164	0.025	−0.039	−0.035	0.832	−0.228	−0.256	0.124	−0.069	−0.016	0.814

Diagonals represent the square root of AVE for each construct, and off-diagonals represent the correlations among constructs. The diagonal elements should be larger than the off-diagonal elements to establish discriminant validity. Note: CET = consumer ethnocentrism, GCV = green consumer, LID = local identity, AUTH = authenticity, HB = healthiness bias, PRI = price consciousness.

**Table 7 foods-10-02715-t007:** Structural model parameter estimates for the seasonal consumption model and for the non-seasonal consumption model (H1–H6).

		Apples (Seasonal)			Tomatoes (Non-Seasonal)		
Hypothesis	Relationship	γ	*t*-Value	Result	γ	*t*-Value	Result
H1	CET → LC	0.268 ***	4.193	supported	0.383 ***	6.666	supported
H2	GCV → LC	0.058	0.898	not supported	0.094	1.547	not supported
H3	AUTH → LC	0.130 **	2.068	supported	0.049	0.882	not supported
H4	LID → LC	0.015	0.225	not supported	0.165 **	2.871	supported
H5	HB → LC	0.193 **	3.122	supported	0.182 ***	3.236	supported
H6	PRI → LC	−0.306 ***	−4.533	supported	−0.429 ***	−6.952	supported
control var.	GAT → LC	−0.092	−1.353	no influence	0.005	0.084	no influence
control var.	GID → LC	0.095	1.452	no influence	−0.040	−0.707	no influence

** *p* < 0.05, *** *p* < 0.001, Note: LC = Local and (non)local seasonal food choice, CET = consumer ethnocentrism, GCV = green consumer, LID = local identity, AUTH = authenticity, HB = healthiness bias, PRI = price consciousness, GAT = globalization attitude, GID = global identity.

## Data Availability

The data presented in this study are available on request from the corresponding author. The data are not publicly available to guarantee maximum data security and privacy of respondents.

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
