# Peer review of "What Drives the Choice of Local Seasonal Food? Analysis of the Importance of Different Key Motives"

_foods, 2021, doi:10.3390/foods10112715_

Round 1

Reviewer 1 Report

The paper is about the consumer’s drivers to choose seasonal food items.  The study is exciting and well-conducted. I found some parts very long; on the other hand, the SEM needs more explanation. I’ve made some suggestions aiming to improve the manuscript, I hope you find them helpful.

Title: I think the title is very long. You are losing impact with such a long title. I suggest you remove “by Means of a Discrete Choice Experiment and Structural Equation Modeling”

Introduction: I found the introduction very long. Since you have a well-described hypothesis section, I think you should include two (max three) paragraphs stating the study's importance and novelty. You use 41 references just for this section. I think it is too much.

L132 – It depends. In many countries, consumer’s are not aware of where the food comes (e.g. south American countries such as Brazil). However, this is an important aspect for food choice in developed countries as EU-countries and US. Maybe you could clarify this aspect.

L373 – Please include the percentage of each characteristic (quota) (or mention table 2)

I’m confused about the sample explanation. 499 seems adequate, but I missed more description about why you chose this sample. For example, you expected high communalities? You have a reasonable number of indicators and split your sample. Are you sure your sample is sufficient? I think you have to give the reader more details about it.

Table 1 – change comma for dot (price and bag)

L436 – Why you used bot cronbach’s alpha and composite reliability? I suggest removing alpha since it is redundant and due to the tau-equivalence assumption.

L462 – You have to justify the model selection. Lisrel uses a covariance-based method? Would you please justify your choice for this model? You developed a new model, I think Partial Least Square would be more adequate.

Did you check data for common-method bias?

Did you check data for collinearity?

Please include the full details of the SEM and factor analysis. Have you conducted an exploratory analysis first?

Table 5 – I suggest removing the cronbach’s alpha

L545 – You have to include this information in the paper (CFI, RMSEA, TFI etc.) What you considered adequate?

Table 7 – Include all acronyms as a footnote. Tables must stand by themselves.

The discussion is adequate.

The authors are very honest in their discussion and limitations. I congratulate you on that.

Author Response

Dear reviewer,

thank you very much for your valuable suggestions. We hope we could address all issues, please refer to pdf-document for our reply. We found your suggestions extremely helpful and think we could significantly improve the manuscript.

Thanks again.

Kind regards

Reviewer 2 Report

There are several issues to be considered and revised in this journal as follows:

*[Abstract] Ln 23: Please suggest additional description about the further research (further research regarding what?).

*[Keywords] Ln 26: consumer preferences -> consumer preference

*Ln 48: communication strategies (communicatin strategies regarding what?)

* [Introduction] Previous relevant researches using choice-based conjoint (CBC) analysis as the major study method are recommended to be introduced in the 3rd and 4th paragraphs of "INTRODUCTION" section.

*Ln 177: Abbreviation "CoO" was already used in Ln 133.

*Ln 185: What is the meaning of (p.280)? This expression is repeatedly indicated in the following contents (e.g., Ln 213) and the authors should provide additional information to understand the readers.

*There are typographical errors in Table 1 to describe the price (the use of ',' for decimal point).

*Ln 641: Please check the expression "which they relate to with their" with the perspectives to the grammar to determine the necessity of text editing.

*Ln 646: Please suggest additional explanation about the "previous findings [18,22–24,74]".

*Please check the typographical errors in the REFERENCE lists.

*Overall manuscript repeatedly used both the abbreviation and the full name of that abbreviation. After the first use of the abbreviation, the indication of the full name of the abbreviation is recommended to be avoided (e.g., Consumer ethnocentrism and CET).

*The contents of "CONCLUSION" section are recommended to be more briefly revised.

Author Response

(The authors gave the same response as above.)

Round 2

Reviewer 2 Report

All comments were applied to revise the manuscript. 

However, this reviewer request the author to check the issue described below:

*Ln 64: Please check whether there is a grammar error or not in the statement "resulting in the of use environmental"